# IS MEMORIZATION *Actually* NECESSARY FOR GENERALIZATION?

## ABSTRACT

Memorization is the ability of deep models to associate training data with seemingly random labels. Even though memorization may not align with models' ability to generalize, recent work by Feldman and Zhang (2020) has demonstrated that memorization is in fact *necessary* for generalization. However, upon closer inspection of this work, we uncover several methodological errors including lack of model convergence, data leakage, and sub-population shift. We show that these errors led to a significant overestimation of memorization's impact on test accuracy (by over five times). After accounting for these errors, we demonstrate that memorization does not impact prediction accuracy the way it is previously reported. In light of these findings, future researchers are encouraged to design better techniques to identify memorized points that can avoid some of the earlier stated problems.

## 1 INTRODUCTION

One of the most interesting properties of deep learning models is their ability to fit outliers (i.e., samples that are not part of the data distribution) (Zhang et al., 2017; Arpit et al., 2017; Stephenson et al., 2021). Specifically, deep models can output arbitrary ground-truth labels to inputs in the data set. For example, if a picture of Gaussian noise is mislabeled as a cat, then the model will output this label, even though the label is incorrect (Zhang et al., 2017). This is only possible due to the model's ability to *memorize* point-label pairs.

Intuitively, the ability to generalize (i.e., correctly label previously unseen points) should be at odds with memorization. This is because generalization requires identifying the underlying patterns and then subsequently applying them to unseen points. On the other hand, memorization simply retrieves the labels of the previously observed inputs and consequently, should not help in correctly classifying new unseen points. However, recent work from Feldman and Zhang (2020) has shown that this is not true for deep models. In fact, their work demonstrated that "memorization is necessary for achieving close-to-optimal generalization error". They show this by 1) identifying the memorized points in the training data, 2) removing these points from the data set, 3) retraining models on the reduced data, and 4) measuring the drop in test accuracy. They report a significant accuracy degradation of 2.54 ± 0.20%, thereby concluding that memorization is necessary for generalization.

While their work makes certain valuable contributions, we show that their conclusion is incorrect. This is due to several methodological errors that include:

- **Lack of Convergence:** Models are not trained to their maximum test set accuracy (Section 4.1).
- **Data Leakage:** Training points have duplicates in the test set, resulting in imprecise test set accuracy (Section 4.2).
- **Sub-population Shift:** Entire sub-populations were removed alongside the memorized points. This gives rise to a widely recognized issue known as sub-population shift (Yang et al., 2023; Santurkar et al., 2021)(Section 4.3).

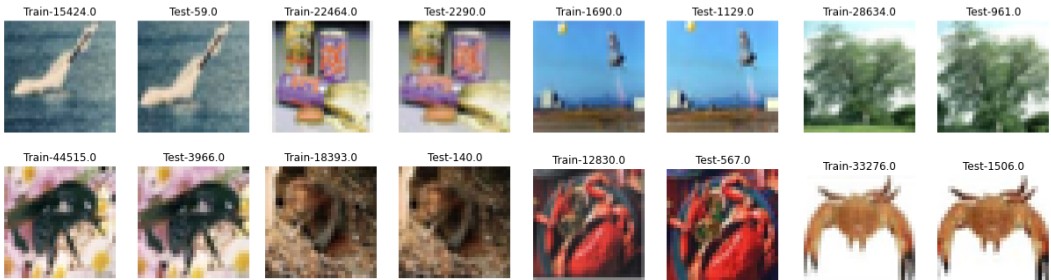

Figure 1: Examples of train-test duplicates present in the CIFAR-100 data set.

In our work, we improve their methodology by accounting for these errors. In doing so, we show that the authors had overestimated the impact of memorization on test set accuracy and that this number is significantly lower. Specifically, while the authors report that removing memorized points leads to a 2.54 ± 0.20% drop in test set accuracy, we demonstrate that this memorization only results in a 0.54 ± 0.29% drop. Below, we describe the high-level steps of our approach.

First, we show that by converging the model to a higher test accuracy significantly reduces the impact of memorization on the test accuracy (by almost a half). Second, we analyze the memorized points and show that almost half of them suffer from data leakage or sub-population shift. To account for them, we clean the memorization data by removing the points that induce this error. Third, we show that even though the remaining memorized points have a *higher* memorization score, they have an insignificant impact on test accuracy. Specifically, the impact of memorized points goes from 2.54 ± 0.20% (as reported by the original paper) to just 0.54 ± 0.29%, which is a five-fold decrease. Therefore, we disprove the conclusion of the original work and show that *memorization is not necessary for generalization*.

Finally, we note that the memorized points, as identified by Feldman and Zhang (2020), exhibit substantial overlap with data points originating from certain sub-populations within the training set. As a result, the role that the memorized points play largely duplicates that of the sub-population in model accuracy. This phenomenon casts doubt on the validity of the original work's definition of memorization in the first place. In light of these results, future researchers are encouraged to put forward more precise definitions of memorization and develop alternative approaches for accurately discerning memorized points. Because ultimately memorized data should not be a redundant representation of sub-populations.

## 2 BACKGROUND

Before we discuss our findings in any detail, it is important that we first understand some of the important background concepts regarding memorization.

### 2.1 DATA LEAKAGE

Data leakage is the use of information by the model at train time that would otherwise not be available to it at test time (Kaufman et al., 2012). An example of data leakage is when the model's training data overlaps with the test one, due to the presence of train-test duplicates. This is when a data point from the training set has an identical point in the test set (Figure 1). This also extends to near identical points as well (Figure 2). Here, two images have identical content (shape, form, background, color, etc), but might have different perspective (angle of the photograph). This is problematic since the goal of the test set is to quantify the model's ability to generalize on *unseen* points. Therefore, there should be no overlap between train/validation and test sets.

Any sort of overlap, e.g., in the form of train-test duplicates leads to an *overly optimistic* and artificially high test set accuracy. This is because the model will correctly classify the duplicates from the test set (as it already saw them during training). The accuracy is overly optimistic as it is a result of train-test duplicates rather than the model's ability to generalize to unseen data. Unfortunately, commonly used data sets in the current machine learning literature (e.g., CIFAR and MNIST) suffer from this issue (Recht et al., 2018).

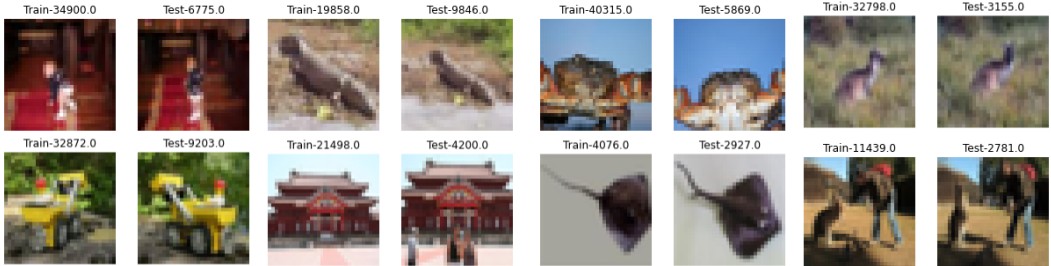

Figure 2: Examples of train-test **near**-duplicates present in the CIFAR-100 data set.

## 2.2 SUB-POPULATIONS

A data set can consist of one or more coarse class labels (e.g., cats and dogs). Within each of these coarse labels, there may exist a mixture of points that have finer labels, which correspond to distinct sub-populations (Zhu et al., 2014). For example, the cat data set will contain images with different cat features including color, background, species, pose, etc. Cats with the same facets will fall into the same sub-populations. For example, consider a hypothetical data set that contains 100 cat images, with 95 white cats and 5 black ones. Even though they have the same label, the white and black cats form two distinct sub-populations (with potentially even finer sub-populations within the white and black cats respectively (Malisiewicz et al., 2011; Felzenszwalb et al., 2009)). Figure 3 provides an example of points from the same sub-populations in CIFAR-100[1]. As we can see, images have similar visual features.

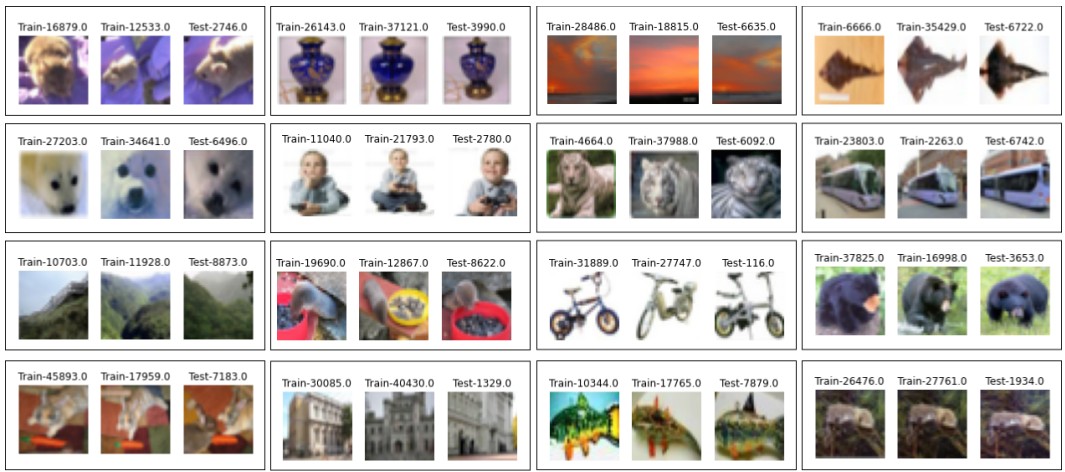

Figure 3: Examples of sub-populations that were removed by Feldman and Zhang (2020) alongside the memorized points and the test set points that were impacted.

The size of sub-population may impact model accuracy as well. Generally, the larger the sub-population, the higher the number of exemplar points, the greater the model's ability to predict accurately for that sub-population at test time. This is because more points usually mean more representative examples for the model to learn from. Returning to our hypothetical cat data set (with 95 white cats and 5 black ones), since there are more pictures of white cats in the data set, the model likely has better prediction accuracy on white cats than black ones at test time. This also means removing an entire sub-population will result in degrading the model's ability to correctly classify missing sub-population at test time. If the model was never trained on black cats, it will likely misclassify them at a higher rate. This is because certain distinguishing characteristics that aid the model in correctly classifying the unseen sub population will likely not be learnt from remaining data set, and will negatively impact the performance on unseen sub-populations.

---

[1]In Section 5 we describe how we find the sub-populations within a dataset.

Table 1: Symbols used and their meanings.

| Symbol | Meaning |
|---|---|
| $x_i$ | training data point |
| $y_i$ | training point label |
| $x'_i$ | test data point |
| $y'_i$ | test point label |
| $S$ | training set |
| $A$ | training algorithm |
| $n$ | size of the training set |
| $m$ | number of points removed from the training set |
| $h$ | trained model |
| $t$ | trial |

## 2.3 Influence of Data Set Points

Influence is the ability of a training point $(x_i, y_i)$ to impact the prediction of some other point $(x_j, y_j)$ (Koh and Liang, 2017). Hereinafter, $x_i$ denotes the data point and $y_i$ denotes the label of $x_i$. As an illustrative example for influence, if including $x_i$ in the training set increases the probability of point $x_j$ being classified correctly by the resulting model, then $x_i$ is said to have a *positive* influence on $x_j$ (Feldman and Zhang, 2020). The higher the influence value, the stronger the impact.

*Self-influence* is a special case of influence. This is the measure of how much $x_i$ impacts the models prediction on itself. In other words, how well the model predicts the label for $x_i$ when the point itself is present in the training data set in comparison to when $x_i$ is absent. If a point has positive self-influence, it has a higher probability of being classified correctly when it is present in the training data set. Therefore, when the point is removed from the training data set, the likelihood of correct prediction goes down as well. Conversely, negative self-influence means a higher likelihood of being classified correctly only if it is not present in the training data set (e.g., mislabeled points).

According to Feldman and Zhang (2020), higher self-influence means a higher risk of memorization. The high self-influence points usually belong to the tail end of the data distribution. The tail usually consists of atypical points (e.g., outliers and mislabeled points) or small-sized sub-populations (e.g., five black cats in a data set of all white cats). Therefore, these points have the highest risk of memorization across the entire distribution.

Furthermore, if the point has high self-influence *and* has a duplicate in the test set, then removing this point from the training data will result in the wrong prediction on itself, but also its duplicate (or near duplicate) in the test set.

## 3 Understanding Feldman and Zhang (2020)

Having gone over how different factors influence memorization, we describe in detail the original work of Feldman and Zhang (2020). Our primary goal is to evaluate their methodology, recommend experimental fixes, and consequently, reassess their findings. To that end, we attempt to gain a deeper understanding of the original work. In this section, we describe how they 1) define memorization, 2) identify memorized points, and 3) quantify their marginal utility.

### 3.1 Defining Memorization

Feldman and Zhang (2020) define a memorized point as one having high self-influence (i.e., a point that is predicted correctly only when present in the training data).

Specifically, consider a training set $S = ((x_1, y_1)...(x_n, y_n))$ and a point $x_i$ in the training set $S$. The memorization score is the difference in prediction accuracy between when the point $x_i$ is present in the training data ($h \leftarrow A(S)$) and when $x_i$ is absent ($h \leftarrow A(S^{\backslash i})$). Here, ($h \leftarrow A(S)$) means that models $h$ were trained on dataset $S$ using algorithm $A$. We include Table 1 for reference on the symbols used throughout the paper:

$$\text{mem}(A, S, i) = \mathbf{Pr}_{h \leftarrow A(S)}[h(x_i) = y_i] - \mathbf{Pr}_{h \leftarrow A(S^{\backslash i})}[h(x_i) = y_i] \qquad (1)$$

The definition captures the intuition that a point $x_i$ has been memorized if its prediction changes significantly when it is removed from the dataset.

For example, consider training 1000 instances each of the models $h \leftarrow A(S)$ and $h \leftarrow A(S^{\setminus i})$. If the correct classification rate for $x_i$ when it $h \leftarrow A(S)$ is around 90% (i.e., 900 out of the 1000 instances classified the point correctly). However, it falls significantly when $h \leftarrow A(S^{\setminus i})$ to 25% (i.e., 250 out of the 1000 instances classified the point correctly). Due to the significant drop in self accuracy, this point has a high self-influence, and therefore, a high memorization score, specifically of $90\% - 25\% = 65\%$. This means that $x_i$ is far more likely be classified correctly when it is present in the training data. In contrast, if there is no significant change in the classification rate, then it has a low memorization score. In this case, $x_i$ will likely be classified correctly, whether or not it is present in the training data.

### 3.2 IDENTIFYING MEMORIZED POINTS

Now that we have defined memorization, the next step is to develop a methodology to identify memorized points from a dataset. A point is considered memorized based on its memorization score, calculated using Equation 1. One way to compute this score is via the classic leave-one-out experiment. Here, we remove a single point from the training dataset, retrain the model on the remaining data, and test to see if the removed point is correctly classified. We have to run this experiment on all the points in the dataset to get the memorization score for each. Additionally, we have to repeat this model training process, for each point, multiple times to account for different sources of randomness introduced during training (e.g., the varying initialization, GPU randomness, etc.). Specifically, this would require training hundreds models for every point in the training data. Considering training data sets contain tens of thousands of points, this would require training millions of models. Therefore, running this experiment over a large dataset and model will require a large amount of resources and is therefore, computationally intractable.

To overcome this limitation, Feldman and Zhang (2020) propose a method to *approximate* the memorization scores. Instead of removing one point at a time, the authors randomly sample a fraction $r$ of the points from the training set (originally of size $n$) and leave the remaining points out of training. The number of points used in training is then $m = r \cdot n$, $0 \leq r \leq 1$. In Feldman and Zhang (2020) the authors use $r = 0.7$ for their experiments. The authors repeat this $k$ times. The exact value of $k$ depends on the dataset but is typically on the order of a few thousand models. As a result, a random point $x_i$ will be present in approximately $k \cdot r$ of the total trained models and will be absent from $k \cdot (1 - r)$ of them. By aggregating the results over both sets of models, the authors can approximate the memorization score for $x_i$. All the points that have a higher memorization score than some predetermined threshold (specified in the original work as 25%) are said to be memorized.

### 3.3 CALCULATING MARGINAL UTILITY

Having identified the memorized points, the authors now calculate their marginal utility (i.e., their impact on test accuracy). This is done using a two-step process:

#### 3.3.1 STEP 1: TRAINING MODELS WITHOUT THE MEMORIZED POINTS

The authors train two sets of models: one on the full training data (that includes the memorized points), and another on the reduced dataset (without the memorized points). They train both sets of models on identical parameters, repeating this training procedure hundreds of times to account for different sources of randomness. At this point, the authors have hundreds of models trained on the full data set and reduced datasets.

#### 3.3.2 STEP 2: MEASURING THE DIFFERENCE IN ACCURACY

Next, the authors measure the drop in accuracy caused by removing the memorized points and retraining the models. They simply take the mean test set accuracy of the models trained on the full dataset and the models trained on the reduced one respectively. They subtract the two accuracies to find the mean difference and the standard deviation. The authors reported a significant drop in accuracy of 2.54 ± 0.20% and therefore, the concluded that these memorized points need to be

present in the training data for optimum accuracy. And as a result, memorization is necessary for generalization.

## 4 GAPS AND FIXES FOR THE EXISTING APPROACH

In the previous section, we describe how Feldman and Zhang (2020) define memorization, identify memorized points, and calculate their marginal utility. However, there are a number of methodological errors in their work, which lead to their incorrect conclusion about memorization. In this section, we describe these errors, and propose experimental fixes:

### 4.1 ERROR: LACK OF MODEL CONVERGENCE TO MAXIMUM TEST ACCURACY

Feldman and Zhang (2020) use non-ideal training parameters which result in the sub-optimal test set accuracy. This is because these parameters do not allow models to learn all necessary patterns from the training set. This issue is far more evident in the case of smaller sub-populations. Here, models have difficulty learning the patterns due to the limited number of points. As a result, removing a single point will significantly impact the accuracy of its own sub-population in the test. Therefore, the decrease in accuracy between the full and reduced data models was not due to the removal of the memorized points. Instead, it was because models had not been trained to learn some of the less obvious patterns.

Fix: One simple fix is to improve the training procedure, allowing models to learn less obvious patterns more effectively, while simultaneously reducing their sensitivity to the removal of any individual point. One popular method of achieving this goal is weight decay (Krogh and Hertz, 1991). This regularization method reduces models' sensitivity to individual samples and improves model generalization. We use this method in conjunction with the original methodology to improve model training.

### 4.2 ERROR: DATASET LEAKAGE

As discussed in Section 2.1, train-test duplicates result in an overly-optimistic and artificially high test set accuracy. This is because the model will correctly classify points whose duplicates it saw during training, even if it performs poorly on the remaining test set. The authors did not account for this behavior. While they kept the duplicates when training the full data models, they removed duplicates as part of the memorized points to train the reduced data models. Therefore, the resulting difference in accuracy was not entirely due to the marginal utility of the memorized points. Instead, it is in part attributed to the removal of duplicates from one of the two training data sets. This led to an unfair comparison between the two resulting models.

Fix: Only remove the memorized points *not* associated with the train-test duplicates. This way, we can measure memorization's impact as if there was no data leakage.

### 4.3 ERROR: SUB-POPULATION SHIFT

When calculating marginal utility, the authors remove memorized points to measure their impact on the test set (Section 3.3.1). A number of these were images from the same sub-population, as can be seen in Fig 3. Since these sub-population images have high self-influence values, this means a small number of images make up these sub-populations. Removing a few points from the already small sub-population will cause a complete or near complete sub-population purge from the training data, and will result in a distribution shift. This is because the training data will no longer possess certain sub-populations that do exist in the test data.

This will prevent the model from effectively learning distinguishing features of the sub-populations (since they were removed from the training data) and consequently, lead to poor model performance on the corresponding test set points.

This idea can be understood using our earlier cat dataset example, presented in Section 2.2. If we remove all the black cats (five of them in total) from the dataset that contains another 95 white ones, the corresponding model will likely perform poorly on black cats in the test set. This degradation will be even more severe if the black cats have high self-influence. This implies that few distinguishing features can be learned from the white cats to help identify the black ones. Therefore,

the drop in model accuracy observed by Feldman and Zhang (2020) was *not* due to the marginal utility of the memorized point but can be attributed to the induced distribution shift (because entire sub-populations had been removed from the training set).

Fix: Only remove the memorized points *not* belonging to any sub-populations. Now, we can study the impact of memorization as if there was no sub-population shift.

## 5    EXPERIMENTAL SETUP AND RESULTS

Having identified the methodological errors in the previous section, we now implement the necessary fixes, rerun the experiments, and report our results.

### 5.1    SETUP

In order to perform the most fair evaluation, we employ an exact same experimental setup as the one used by Feldman and Zhang (2020). Specifically, we train Resnet50 (He et al., 2016) models on the full CIFAR-100 data sets for 160 epochs, using a batch size of 512, momentum of 0.9, and triangular learning rate scheduler, with a base rate of 0.4. However, to account for the lack of convergence (Section 4.1), we use weight-decay. This helps the model learn better patterns and improve test time accuracy. We train over 100 models in each setup and use the FFCV library (Leclerc et al., 2023) to quickly train a large number of models.

Next, we remove the same 1,015 memorized points provided by the original authors and retrain the models. We do this for two reasons: 1) We want to recreate the exact same setup as the original authors for the most fair comparison. 2) These 1,015 points are the ones that had the highest impact on the test set accuracy. Refuting the authors' claim against the most impactful memorized points means that the impact of other memorized points is automatically dismissed.

Now, we address the data leakage error (Section 4.2). We iterate over the removed memorized points and find their exact or near duplicates in the test set (Figure 1). We were able to identify 119 duplicates and 158 near duplicates.

Similarly, we account for the sub-population shift (Section 4.3). Here, we identify the sub-populations based on the observation from prior work that the accuracy of models on a sub-population is likely to increase once a representative example from that sub-population is observed during training (Feldman, 2020). In other words, the presence of data points from a specific sub-population in the training data tends to enhance models' accuracy on other points from the *same* sub-population in the test set. For instance, in our cat dataset example, including black cat images in the training set can lead to improved model accuracy on black cat images in the test set. Following this rationale, we perform a two-step process to identify sub-populations: 1) check if multiple memorized points impact the same test set point. If that is true, then they belong to the same sub-population; 2) perform an additional visual check to confirm that these points do belong to the same sub-population. To show the validity of this method, we show an example array of the sub-populations that we found in Figure 3. We identified 239 such points. In total, we remove $119 + 239 + 158 = 516$ points from the original 1,015 points and are left with 499 memorized points.

The goal of our experiments is to understand 1) whether memorized points (barring the errors) have a significant impact on test set accuracy and 2) if data leakage and sub-population shift contribute towards a greater portion of the loss in accuracy (compared to the actual memorized points). To answer these questions, we split the memorized points into two buckets. We place the removed points in the "Leakage+Shift" bucket. And we place the remaining points in the "Memorized" bucket. We remove each bucket from the training data individually train 100 models, and observe the drop in accuracy for each bucket.

### 5.2    RESULTS

We compare the results of our findings against the original work in Table 2. We can see that the originally reported drop in model accuracy (across hundreds of models) after the memorized points are removed is 2.54 ± 0.20%. However, we show that merely training these models to maximum

Table 2: The table shows the impact of memorization points on test set accuracy. The original paper reported a drop of 2.54 ± 0.20%. However, we can see that after training the models to convergence, the value is significantly 1.78 ± 0.32%. We can also see that the majority of the drop was due to data leakage and sub-population shift errors (Leakage+Shift Bucket). However, the Memorized Bucket has an insignificant 0.54 ± 0.29% drop in accuracy, a five-fold decrease from the original number.

| | Original Result | Our Results | | |
|---|---|---|---|---|
| | | Model Convergence | Leakage+Shift Bucket | Memorized Bucket |
| Accuracy Drop | 2.54 ± 0.20% | 1.78 ± 0.32% | 1.25 ± 0.32% | 0.54 ± 0.29% |

test accuracy reduces this value to almost half, at 1.28 ± 0.48%. This shows that *adequate training can reduce memorization points' impact on the test set*.

Next, we measure whether the Leakage+Shift or the Memorized bucket contributes towards a greater portion of the drop in accuracy. We can see in the Table that the Leakage+Shift bucket has more than two times larger impact on test accuracy than that of the memorized bucket (1.25 ± 0.32% vs 0.54 ± 0.29% respectively). The drop in accuracy from 2.54 ± 0.20% as reported in the original paper is five times larger than what is found after accounting for these errors. In other words, the actual accuracy drop is almost exclusively caused by data leakage and sub-population shifts. More importantly, upon closer inspection, we found that the points in the Memorized bucket in fact had a lower memorization score on average than the Leakage+Shift bucket (83% and 74% respectively based on Equation 1). This shows the memorized points had a lower accuracy drop despite a higher memorization score. These results demonstrate the following: 1) Higher memorization scores do not correspond to a higher drop in accuracy. 2) The reduction in test set accuracy in the original paper was due to improper training, data leakage, and sub-population shifts. 3) Finally, memorization has an insignificant impact on test accuracy, and therefore *is not necessary for generalization*

## 6  DISCUSSION

While Feldman and Zhang (2020) made valuable contributions, in the previous section, we showed that overestimated the impact of memorization on test set accuracy by a factor of five. After accounting for the three errors (model convergence, data leakage, and sub-population shift), we show that memorization does not have a significant impact on accuracy. While the errors of model convergence and data leakage are straightforward to deal with (specifically, train models to maximum accuracy and avoid train-test duplicates, respectively), fixing the sub-population shift is not easy. This is because the issue of sub-population shift directly resulted from Feldman and Zhang's definition of memorization and the technique they use to identify memorized points (Section 3). The authors use a memorization (self-influence) threshold of 25%. Any points greater than this threshold are considered memorized. Memorized points produced in this fashion significantly overlap with certain sub-populations, especially those consisting of fewer samples (because these have higher self-influence values). This raises the question: *Do memorized points merely become a redundant representation of the sub-populations?* In light of our results, future researchers are strongly encouraged to put forward a more precise definition of memorization and explore more robust methods for accurately identifying memorized points that are distinct from sub-populations.

Memorization has a direct implication for privacy research. This is because memorized points are vulnerable to membership-inference attacks (Carlini et al., 2022a). Feldman and Zhang (2020) created a tension between generalization and privacy. This is because they claimed that memorization was needed for generalization while other works demonstrated that memorization was harmful to privacy (Carlini et al., 2022a; Leino and Fredrikson, 2020; Carlini et al., 2019; Li et al., 2022). In other words, generalization and privacy can not be simultaneously achieved. While this might have dissuaded researchers in the community, our work shows that this tension does not exist. This is because memorization is not necessary for generalization. In light of these results, future researchers are encouraged to explore methods to build models that both generalize and are private.

## 7 RELATED WORK

One of the first papers to discover memorization deep learning models was Zhang et al. (2017). They showed that models can fit completely unstructured images even if these consist of random Gaussian noise. Since then, there has been a tension between memorization and generalization and how they impact model performance (Chatterjee, 2018). Earlier works focused on limiting model memorization, thereby forcing the model to learn patterns instead. This was partly motivated by the fact that memorization exposed models to privacy risks (e.g., membership inference) (Carlini et al., 2019; 2022b). As a result, different methods were developed to counter memorization, which included using regularization (Arpit et al., 2017), filtering weak gradients (Zielinski et al., 2020; Chatterjee, 2020), adjusting model size (Arpit et al., 2017; Zhang et al., 2019). While these methods did reduce model memorization, they did so at the cost of model accuracy.

However, the true impact of memorization on model behavior was yet unknown. This first and foremost required methods to identify memorized points. A number of post-hoc methods were developed to identify them. These included clustering (Stephenson et al., 2021), repurposed membership inference attacks (Carlini et al., 2022b), pseudo leave-one-out method (Feldman and Zhang, 2020). Having developed the ability to identify these points, the authors were now able to study their impact on model efficacy. As we describe (Section 3), Feldman and Zhang (2020) demonstrated that memorization was in fact necessary for model memorization. However, this conclusion was incorrect and was a by-product of a number of methodological errors. By accounting for these errors and rerunning their experiments, our results show that memorization has minimal impact on generalization.

## 8 CONCLUSION

Memorization is the ability of the model to fit labels to seemingly random samples. Recent work from Feldman and Zhang (2020) demonstrated that memorization is necessary for generalization. We show that the original work suffered from a number of methodological errors including lack of model convergence, data leakage, and sub-population shift. In order to study the real impact of memorization, we modify the original methodology fix the underlying errors, and rerun the original experiments. We show that memorization does not significantly impact memorization. While the lack of model convergence and data leakage are easy to fix, sub-population shifts are harder. This is because the definition of memorization proposed by Feldman and Zhang (2020) and the technique they use to identify memorized points may be flawed in the first place, making memorization largely a redundant concept of sub-populations. In light of these results, researchers and practitioners are encouraged to put forward more precise definitions of memorization and develop alternative approaches for accurately discerning memorized points that are distinct from sub-populations.

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
