## A   APPENDIX

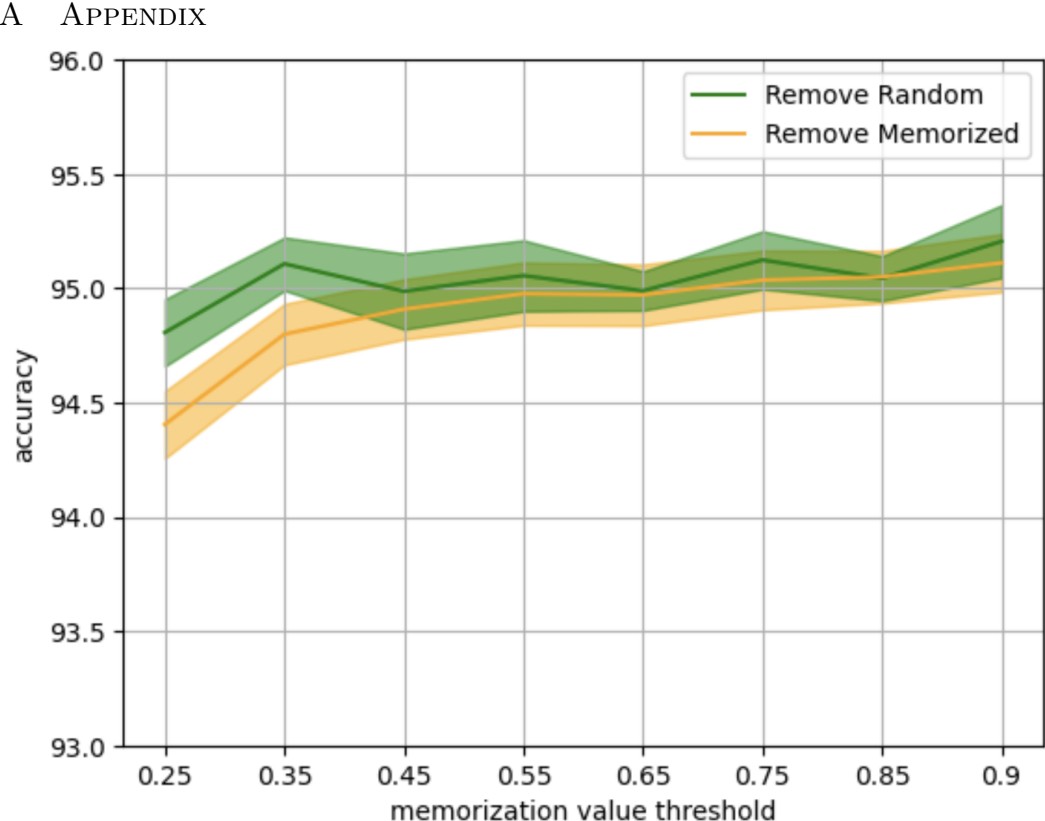

Figure 1: The impact removing examples with memorized points above a given threshold and the same number of randomly chosen examples on the test set accuracy. The plot shows the test set accuracies for models when the memorized points were removed (orange) vs when the same number of memorized points were removed from the training data (green). We can make a few observations: **1.** The difference in accuracies is very small, as can be seen the the y-scale. **2.** Even after removing all the memorized points at the 0.25 threshold (a total of 3,347), there is a less than half a percent drop in accuracy (of $0.4\pm0.2\%$). **3.** For most memorization thresholds, the error bars overlap. This means removing memorized points results in the same approximate drop in accuracy as removing an equal number of random points. This means that the drop in accuracy when removing memorized points is not significant. As a consequence of these three observations, we can conclude memorized points do not significantly contribute towards test accuracy. And therefore, memorization is not necessary for generalization.