# OpenReview forum: "Is Memorization Actually Necessary for Generalization?"
_ICLR.cc/2024/Conference — Submitted to ICLR 2024_

### Official Review · Reviewer_DdXc · 2023-10-31

**Soundness:** 2 fair
**Presentation:** 3 good
**Contribution:** 1 poor
**Rating:** 3
**Confidence:** 4

**Summary:**

This paper challenges the previous finding that memorization is necessary for generalization. They study the same setting as in Feldman and Zhang (2020), except (i) the models are trained to convergence, (ii) data points appearing both in the train and test set are handled properly, and (iii) memorized points that belong to sub-populations are preserved. The results indicate that most of the accuracy drop reported by Feldman and Zhang (2020) can be recovered by the above-mentioned measure.

**Strengths:**

This work touches upon an interesting observation and shows how fragile/unreliably the findings of Feldman and Zhang (2020). Other than this, I do not see any strength in this work.

**Weaknesses:**

- I feel fix-1 changes the entire story. First, "lack of model convergence" sounds like gradient descent hasn't converged, which is slightly misleading. Second and more importantly, it is natural to expect that changing the optimization objective or including different regularization techniques, architectures, etc would impact the findings. After all, the previous work focuses on one particular setup, which would of course make their findings difficult to transfer to new setups.
- Because the setup has changed in certain ways, one may expect different train points to be memorized. Hence, the change in the accuracy should be computed after the points memorized *in this setup* are removed (not the ones reported in the reference paper).
- The analysis is only done for one dataset while the reference paper considers three datasets, which restricts the generality of the findings in this paper.
- More than half of the original memorized points are included in the new training set. In other words, the post-pruning training dataset has a bigger size, which would naturally incur a smaller change in accuracy; hence comparing the reported accuracy change (0.54) against the original number (2.54) would not be fair.

**Questions:**

- _Entire sub-populations were removed alongside the memorized points._ <-- What is a sub-population?
- _original paper reported a drop of $2.54 \pm 0.20%$. However, we can see that after training the models to convergence, the value halved to $1.78 \pm 0.32%$._ <-- Does the number really halved?
- _We can see in the Table that the Leakage+Shift bucket has a three times larger impact on test accuracy than that of the memorized bucket (1.25 vs 0.54)_. <--- Again, is the math correct?

---

> ### Author Response · Authors · 2023-11-21
>
> We would like to thank the reviewer for their comment. They raise really good points.
>
> 1. "After all, the previous work focuses on one particular setup, which would of course make their findings difficult to transfer to new setups"
>
> We agree with the reviewer, that changing setups can change findings. And that is precisely our point. [1] made a general claim that memorization was necessary for generalization. If their claim is valid, changing the specifics of training pipeline (such as adding regularization) should not matter. However, we see that this is not the case. Our results show that using regularization alone reduces the impact of memorization on test accuracy. This means that the notion memorization is necessary for generalization is flawed.
>
> 2. "Because the setup has changed in certain ways, one may expect different train points to be memorized."
>
> We thank the reviewer for this comment. Upon closer inspection, we saw that our setup memorized a strict subset of the points that were memorized by [1]. So our setup did not memorize any new points, just a smaller portion of the original.
>
> 3. "The analysis is only done for one dataset"
>
> We express our gratitude to the reviewer for providing this insightful comment. In response to your suggestion, we conducted additional experiments on the CIFAR-10 dataset to assess the generalizability of our findings, and we are pleased to report that they indeed do generalize!
>
> Following the protocol outlined in Section 5.1, we identified 4,126 memorized points. Among these, 779 are associated with sub-populations and near-duplicates (Leakage+Shift Bucket), while the remaining 3,347 points do not exhibit this issue (Memorized Bucket). Eliminating all 3,347 memorized points from the training data resulted in a marginal accuracy drop of 0.4±0.2\%. In contrast, removing the 779 points that include near-duplicates and sub-populations led to an accuracy decrease of 0.5±0.11\%. This underscores that the removal of points from the Leakage+Shift bucket has a more pronounced impact on test accuracy than the memorized bucket, even though the Memorized Bucket is four times larger than the Leakage+Shift Bucket (3,347 vs 779).
>
> It's worth noting that we did not assess our method against MNIST, as this dataset has few memorized points, as indicated by [1] in the original work. Additionally, due to significant computational demands, we were unable to evaluate our approach on Imagenet during this rebuttal phase. Given these constraints, our evaluation focused on CIFAR-10.
>
> 4. "More than half of the original memorized points are included in the new training set..."
>
> We thank the reviewer for this comment.
> In the CIFAR-100 experiments, even though we only removed half the training points, the drop in accuracy is not at all proportional. The accuracy drop decreased from 2.54 to 0.54, a five-fold decrease.
>
> Furthermore, in our CIFAR-10 experiments, we show that removing _all_ the 3,347 points from the memorized bucket still only incurs a rather small 0.4±0.2\% drop in accuracy (Figure 1 of the appendix.). This means our original findings still hold.
>
> 5. "Does the number really halved?"
>
> Thank you for this suggestion. We have corrected it.
>
> 6. "Again, is the math correct?"
>
> Thank you for this suggestion. We have corrected it.
>
> 7. "What is a sub-population?"
>
> Please refer to Section 2.2.

---

> > ### Comment · Reviewer_DdXc · 2023-11-22
> > **my response to author response**
> >
> > Thanks for the author response. Concerning my main issue; I don't see where/how [1] made the general claim that "all neural networks trained on all datasets using all possible loss functions have to memorize to generalize". Hence, if the reviewed work claims to falsify [1] by a counter-example, which is my understanding of the author's response to my first comment above, then I surely disagree with this. To my understanding, [1] is an attempt to empirically evaluate the claims in the "Does Learning Require Memorization? A Short Tale about a Long Tail" paper. At most, the reviewed work provides a more careful look at the setup considered [1].
> >
> > Further, now re-considering the three fixes, they rely on access to the test set. This makes the whole procedure even more questionable from a conceptual perspective.
> >
> > Finally, I also would like to express my gratitude for conducting one more experiment on a smaller portion of the dataset considered in the main submission. Unfortunately, it does not address my concern.

---

### Official Review · Reviewer_Bqf3 · 2023-11-01

**Soundness:** 1 poor
**Presentation:** 1 poor
**Contribution:** 2 fair
**Rating:** 3
**Confidence:** 4

**Summary:**

This paper investigates the relationship between memorization and generalization in machine learning models. Previous works have conducted empirical studies based on theoretical foundations, demonstrating that memorization is necessary for achieving better generalization on long-tail distributions. However, in this paper, the authors argue that there are flaws in the methodology of these previous studies, which led them to question the necessity of memorization. Therefore, the authors identify these issues and propose potential fixes, ultimately suggesting that memorization may not be necessary for the model to generalize, thereby challenging the previous findings.

**Strengths:**

Despite there being ample room for further improvement of the paper, there are interesting visualizations of memorized and sub-population shifts in the paper.

**Weaknesses:**

In general, the presented study appears to be inadequately explored, leaving more room for further empirical research and justification of observations based on theoretical foundations. The stated issues do not necessarily contradict the findings of the previous work, especially in the case of the "Sub-population Shift" argument, which I will explain further in my review. Therefore, this paper may not be suitable for this particular venue and might be better suited as a workshop paper or considered for the reproducibility track to facilitate further experimentation and validation of the results.

I am strongly leaning towards rejecting the paper; however, I will try to provide my understanding, concerns, and suggestions for the authors so they can reevaluate their work and make a better presentation of their study.

Concerns, understanding, and suggestions:
1. The introduction section does not adequately define the problem, and the background section has not provided sufficient background information.
2. Regarding errors stated in the paper:
    1. “Lack of Convergence: Models”: Why should models be trained to their maximum test set accuracy? You don't have access to the test set during training!
    2. “Data Leakage: Training”: The authors of the original work stated this in their paper in Figure 3 that there is a close match of memorized points in the test set.
    3. "Sub-population Shift": The argument does not necessarily contradict the previous work because it aligns with what they have been arguing in their paper: memorization is necessary due to the long-tail distribution. Both mislabeled and sub-population data points are part of the long tail; therefore. It does not necessarily mean that the model memorized the points associated with sub-populations. They have high influence scores because they are rare in the data distribution, and the model exhibits a similar behavior of memorization.
3. There is an argument in the paper that the definition of memorization from the original work is incorrect, but there is no strong justification for this claim, and it is deferred to future research. I would suggest the authors first review their observations and results and then use them to formulate a more robust definition of memorization. In that case, your work would be better supported.
4. There are many repetitions of information with different formats conveying the same content in the paper, such as in the Introduction, Background, Related work, Discussion, and Conclusion. The presentation format could be significantly improved.
5. The "Data Leakage" in the Background section is not sufficiently well explained. I have read the "Do CIFAR-10 classifiers generalize to CIFAR-10?" paper, and the context provided in your paper is what they have already explained, and it can be moved to the appendix.
6. Table 1: It is not necessary to provide a table for symbols in the main pages; it can be moved to the appendix.
7. A large portion of the paper is dedicated to understanding the original paper, and it is defined in a way that favors the introduced issues mentioned at the beginning of your paper. It can be shortened, and the detailed discussion can be left in the appendix.
8. Section 5.1 (SETUP): You should check what your models memorize with the modifications you have made, not just the 1015 memorized points provided by the original authors.
9. The implementation of fixes is not provided along with the submission.

**Questions:**

Questions:
1. The previous work that you studied is based on a theoretical background. Could you please delve into the details of the theory part and justify your observation? I believe that if your argument is completely true, it should also be possible to find a flaw in the theory of the work.
2. Section 5.1 (SETUP): How did you define which points impact the same test point?
3. Section 4.2 (ERROR: DATASET LEAKAGE): Are you assuming that duplicate points are only found in memorized points?

---

> ### Author Response · Authors · 2023-11-21
>
> We thank the reviewer for the detailed review and appreciate the effort they have put into their review to help improve our work.
>
>
> 1. “Lack of Convergence: Models...”: Why should models be trained to their maximum test set accuracy? You don't have access to the test set during training!
>
> Thank you for pointing this out. We had meant the validation set. We have fixed this in the paper.
>
> 2. “Data Leakage: Training...”: The authors of the original work stated in their paper in Figure 3 that there is a close match of memorized points in the test set.
>
> Yes, they did state that they found close matches. However, they should not have included those points in the final results, which resulted in data leakage and skewed their findings. Preventing data-leakage is one of the most fundamental aspects of machine learning, and therefore has to be avoided when evaluating models.
>
> 3. "Sub-population Shift: The argument..."
>
> We appreciate the reviewer's input. The key idea we aimed to convey is that regardless of whether a set of points is memorized, eliminating entire sub-populations from the training data will lead models to misclassify test data from the same sub-population. This aligns with a fundamental principle in machine learning: models need exposure to data from the same distribution as the one they are tasked with predicting.
>
> To illustrate, suppose there is a dataset with 95 white cats and 5 black cats. Whether the model memorizes information about black or white cats, removing the entire sub-population of black (or white) cats will result in decreased accuracy, as the model will misclassify black (or white) cats during testing. Therefore, the decline in accuracy is not solely attributable to memorization but rather to the removal of entire sub-populations.
>
> Our research demonstrates that Feldman and Zhang's approach captures numerous sub-populations, not just the memorized points. Consequently, when they eliminate both the sub-populations and the memorized points, a decrease in accuracy is observed.
>
>
> 4. "There is an argument in the paper..."
>
> Thank you for this comment. Our argument is an extension of the above response. Since the threshold to define memorization (set at 0.25) is arbitrary, and it captures a significant number of sub-populations. Does it mean that the captured points are even memorized? While reformulating the definition from memorization from scratch is definitely an interesting problem, our goal is to show that even if we accept the current definition, the final findings are not correct (i.e., memorization is necessary for generalization).

---

> > ### Comment · Reviewer_Bqf3 · 2023-12-04
> >
> > Thank you to the author for providing their responses; however, I have outlined my concerns/questions in my original review, and your responses do not address them. Therefore, I must stick to my original score.

---

### Official Review · Reviewer_Uoom · 2023-11-02

**Soundness:** 2 fair
**Presentation:** 4 excellent
**Contribution:** 3 good
**Rating:** 3
**Confidence:** 5

**Summary:**

<This paper does not follow the ICLR format>

The paper in question conducts a thorough examination of the claims made by Feldman and Zhang regarding the necessity of memorization for optimal generalization. The authors of the reviewed work identify three core issues that they believe led to overstated results in the original study: lack of model convergence, data leakage, and subpopulation shift. They address these issues by implementing weight decay for better model convergence, removing near-duplicates from datasets, and conducting a subpopulation analysis to preserve unique subpopulation examples. Their findings suggest that by addressing these factors, the impact of memorization on accuracy is significantly less than previously reported, reducing the loss in accuracy due to memorization by a factor of five.

**Strengths:**

1. The paper is well-structured, offering a clear foundational context for the issues tackled.
2. Figures accompanying the explanations provide clarity and enhance understanding of the exact problem that the authors identify in the previous work.
3. The results challenge the widespread belief about memorization's role in generalization and show a surprising reduction in accuracy loss, suggesting that previous results were indeed overstated.
4. I particularly  like the results on model convergence and data leakage where it is sensible to remove near duplicates.

**Weaknesses:**

<This paper is not following the ICLR format>

1. The critique regarding subpopulation shifts questions the validity of ignoring the central hypothesis of Feldman and Zhang's work, which emphasized the importance of memorizing near-singleton subpopulations. In particular, when you put examples from small subpopulations back into the original training set in order to allow for the test set examples from the same subpopulation to perform well, that defeats the entire purpose of the original paper where they claim that because some test set subpopulations have only one or two members in the training set, therefore they need to be memorized to perform well. So the results on the subpopulation shift are unfounded in my opinion, but I do agree with the results on model convergence and data leakage particularly, but I do not think that the effect is as prominent as the original claim of the paper.
2. The paper fails to replicate a key analysis from Feldman and Zhang's work, specifically the varied levels of model accuracy against memorization thresholds, limiting the robustness of the current findings.  I think that this work needs to do a proper analysis to replicate figure 2 of the paper by Feldman and Zhang where they have various levels of model accuracy with respect to memorization value threshold on ImageNet CIFAR-100 and MNIST datasets, and they show that removing random examples is better than removing memorized examples. In my opinion, this is a critical figure and a critical analysis that the authors need to present, whereas at this point, the authors have only presented results at one single point, which does not account for a solid justification that a prior method does not succeed.
3. There is a concern about the fair comparison due to the reduction of examples used in the training set without adequately accounting for this change in the experimental setup. (See Questions)

**Questions:**

1. Would the accuracy loss remain lower if an additional 516 random data points replaced the omitted 499 examples, keeping the training set size constant?

2. Can the authors replicate the original paper's results across different memorization thresholds, particularly on the CIFAR-100 dataset, and possibly on other datasets as well? This would provide a more comprehensive view of the memorization effects at varying levels.

---

> ### Author Response · Authors · 2023-11-20
>
> We would like to thank the reviewer for the comments and questions.
> 1. "The critique regarding subpopulation shifts..."
>
> We thank the reviewer for this comment. The point we were trying to make is, whether or not a group of points is memorized, removing entire sub-populations from training data will cause models to mis-classify the test data from the same sub-population. This is in fact one of the most fundamental principles of machine learning: models must have observed data from the same distribution of those that they are asked to predict.
>
> For example, consider a dataset of 95 white cats and 5 black ones. Whether black cats or white cats are memorized, removing the entire sub-population of black (or white) cats will result in a drop in accuracy as models will misclassify the black (or white) cats at test-time. So the drop in accuracy can not be attributed to memorization, but instead, the removal of entire sub-populations.
>
> And as we show in our work, Feldman and Zhang's method captures a large number of sub-populations, in addition to the memorized points. Therefore, when they remove the sub-populations, alongside the memorized points, they observe a drop in accuracy.
>
> 2. "The paper fails to replicate a key analysis from Feldman and Zhang's work"
>
> We want to thank the reviewer for this comment. We followed your suggestion and recreated the experiment across different thresholds on CIFAR-10. We used this data set for two reasons. 1) to show that our results are generalizable 2) Because CIFAR-10 models are faster to train than CIFAR-100 models.
> We show our results below:
> | Threshold | Remove Random (Accuracy %) | Remove Memorized (Accuracy %) | Difference (Accuracy %) |
> |:---------:|:--------------------------:|:-----------------------------:|:-----------------------:|
> |    0.25   |         94.81±0.15         |           94.4±0.15           |        0.4±0.2        |
> |    0.35   |         95.06±0.15         |           94.8±0.13           |        0.29±0.14        |
> |    0.45   |         95.08±0.17         |           94.91±0.13          |         0.15±0.2        |
> |    0.55   |         95.07±0.13         |           94.98±0.14          |        0.14±0.17        |
> |    0.65   |         95.03±0.13         |           94.97±0.13          |        0.07±0.22        |
> |    0.75   |         95.13±0.11         |           95.04±0.13          |        0.11±0.15        |
> |    0.85   |         95.02±0.13         |           95.05±0.11          |        -0.09±0.13       |
> |    0.90   |         95.16±0.17         |           95.11±0.13          |        0.05±0.23        |
>
> The Table shows the test set accuracies for models when the memorized points (Memorized Bucket from Table 2 that does not suffer from sub-population and near-duplicate issues) were removed, when the same number of random points are removed from the training data, and the mean difference (with standard deviation) between the two. We can make a few observations:
> 1. The difference in accuracies is very small.
> 2. Even after removing all the memorized points at the 0.25 threshold (a total of 3,347), there is a less than half a percent drop accuracy (of 0.4±0.2\%).
> 3. For most memorization thresholds above 0.35, the mean difference is smaller than the standard deviation. This means that drop in accuracy when removing memorized points is not statistically significant.
>
> As a consequence of these three observations, we can conclude memorized points do not significantly contribute towards test accuracy. And therefore, memorization is not necessary for generalization. We encourage the reviewer to refer to our results in the newly added Figure 4 (in the Supplementary Material) of the paper.
>
>
>
> 3. "Would the accuracy loss remain lower if an additional 516 random data points replaced the omitted 499 examples, keeping the training set size constant?"
>
> We thank the reviewer for this comment. We follow the same methodology laid out in the original Feldman and Zhang's work so that it's an apple-to-apple comparison. Specifically, we always compare a model trained on a reduced dataset (with memorized points removed) with another model with an equal number of random points removed as well.
>
> Finally, thank you for identifying the issue with ICLR format. We fixed it and this saved us a few lines :)

---

### Official Review · Reviewer_9ksD · 2023-11-04

**Soundness:** 3 good
**Presentation:** 4 excellent
**Contribution:** 3 good
**Rating:** 6
**Confidence:** 3

**Summary:**

_Note: the paper seems to be using a different font (font weight? I'm not sure) than the one provided in the conference style file, but this font seems extremely close and to just take slightly more space, so I doubt the authors are using it to "cheat". Also it's maybe just my PDF reader having a moment._

This paper is very much a follow up of [1], and points out 3 methodological mistakes done by the original study.
- The lack of weight decay
- Near-duplicates found in both train and test
- Sub-population removals due to the way points were counted as memorized

By correcting for these, the authors show that the effect measured by [1], while still present, is much more marginal.

[1] V. Feldman and C. Zhang. What neural networks memorize and why: Discovering the long tail via influence estimation. Advances in Neural Information Processing Systems, 2020.

**Strengths:**

This is good science, the paper doesn't go into a lot of depth other than the experiments it does and the reasoning for them, but it paints a complete picture, and rectifies past work.

**Weaknesses:**

This is in some sense very minimal a contribution. It's not quite a negative result because it seems like the effect, while lessened, still exists. There's also no novel algorithmic contribution, e.g. finding better measures of memorization is left to future work. Most of the methodology stems from prior work (other than the corrected mistakes of course).

Because of this, I feel like I don't have much to say in this review. This is good work but also really just above what I'd consider sufficient for a conference paper.

**Questions:**

- "and therefore, [memorization] is not necessary for generalization." I'm curious why the authors claim this if the effect still exists. Is it not statistically significant away from being 0? I would rephrase the abstract to be a bit more conservative.
- I didn't find in [1] nor in this paper how training accounts for the removal of data, i.e., in Table 2, are all models trained on the same number of points? Doing otherwise would seem... incorrect?
- I understand the immense computational costs here, but [1] runs these experiments on 3 datasets (MNIST, CIFAR100, & ImageNet). How do we know if these results are generalizable? (A cheap way would be to confirm this on synthetic data with artificial problems like subpopulations).

---

> ### Author Response · Authors · 2023-11-20
>
> We thank the reviewer for the comments. Before we start the response, we want to point out that we performed the additional experiments that the reviewer had requested, and results show that our findings still hold. We provide the details in our response below.
>
> 1. "I would rephrase the abstract to be a bit more conservative."
>
>     Thank you for the suggestion. We reworded the abstract as per your request.
>
> 2. "I didn't find in [1] nor in this paper how training accounts for the removal of data..."
>
>     In Table 2, the models in the first two columns (Original Result and Model Convergence) are trained on exactly the \textit{same} points. Specifically, around 1,000 points were removed from the dataset. The change in test accuracy (due to memorization removal) goes down from 2.54 ± 0.20\% (first column) to 1.78 ± 0.32\% (second column). This shows that training models to convergence alone significantly reduces the impact of memorization on the test set (i.e., memorization decreases while generalization increases). Similarly, the last two columns (Leakage+Shift Bucket and Memorized Bucket) were trained on approximately the same number of points, with 500 points removed. This shows that the removal of sub-populations and duplicates from the training data was responsible for the drop in accuracy, instead of the memorized points themselves.
>
> 3. "How do we know if these results are generalizable?"
>
>     We would like to thank the reviewer for this comment. We followed your suggestion and ran additional experiments on the CIFAR-10 dataset to see whether our results would  generalize or not. And we find that they do!
>
>     We used the setup as defined in Section 5.1. We identified 4,126 memorized points. 779 belong to sub-populations and near-duplicates (Leakage+Shift Bucket). While the 3,347 points do not have this problem (Memorized Bucket). Removing all of these 3,347 memorized points from the training data results in a rather minor drop in accuracy of 0.4±0.2\%. On the other hand, removing the 779 points consisting of near-duplicates and sub-populations results in an accuracy drop of 0.5±0.11\%. This again shows that the much smaller Leakage+Shift Bucket has a significantly larger impact on test accuracy than the four times large Memorized Bucket (779 vs 3,347).
>
>     We did not evaluate against MNIST since this dataset does not have many memorized points (as pointed out by [1] in the original work). Similarly, due to the immense computational costs, we could not evaluate our method on Imagenet within this rebuttal phase. In light of these two restrictions, we evaluated our work on CIFAR-10.
>
> Lastly, thank you for pointing out the issue with the ICLR format. We fixed it and this saved us a few lines :)

---

> > ### Comment · Reviewer_9ksD · 2023-11-22
> >
> > Thanks for the explanations.
> >
> > >  trained on approximately the same number of points, with 500 points removed
> >
> > I think it would be good to be very clear about this, and ideally have an experiment with the _exact_ same number of points. 500 points may seem like very little but considering CIFAR-10/100 consist of 50 000 examples, that's 1% of the data -- a non-trivial amount.

---

> > > ### Author Response · Authors · 2023-11-23
> > >
> > > Thank you for your comment.
> > > We understand your message to mean one of two things:
> > > 1. Each bucket (Memorized and Leakage+Shift) should contain exactly 500 points:
> > >
> > > While the Memorized Bucket contained 499 points and the Leakage+Shift Bucket contains 516 points (Pg 7). Since the difference in the number of points from each bucket is very small (499 vs 516), splitting each bucket to contain equal number of points will not make a significant difference in accuracies. Therefore, our results will not change.
> > >
> > > 2. We should run an experiment with 500 non-memorized points removed (i.e., points with the memorization score<0.25):
> > >
> > > We actually ran that experiment as well. However, since these points had a low self-influence, they barely had any impact on the test set accuracy. Therefore, we did not report it in this work.
> > >
> > > We would appreciate it if you could provide clarification.

---

> > > > ### Comment · Reviewer_9ksD · 2023-11-23
> > > >
> > > > I meant option 2, and I'm happy to learn that you've run this and that it has barely any impact.

---

### Author Response · Authors · 2023-11-21
**Paper Edits:**

1. New experiments on CIFAR-10 to show generalizability. (Reviewer 1,2,4)
2. Part of Abstract reworded. (Reviewer 1)
3. ICLR Format fixed. (Reviewer 1,2)
4. Removed accuracy decreased by half (2.54 vs 1.78). (Reviewer 4)
5. Removed accuracy decreased by third (1.25 vs 0.54). (Reviewer 4)
6. Replaced "maximum test set accuracy" with "maximum validation set accuracy". (Reviewer 3)

---

### Meta-Review · Area_Chair_kBi4 · 2023-12-11

**Metareview:**

This paper revisits the work of Feldman and Zhang (2020) which posits that natural data distributions are often long-tailed with rare and atypical examples, which deep learning models tend to memorize in order to achieve close-to-optimal generalization. This paper brings up 3 methodological concerns with the original study:  (a) The lack of model convergence, (b) near-duplicates found in both train and test, and (c)  sub-population shifts. They conclude that once these are addressed, the impact of memorization on generalization seems diminished.

**Justification For Why Not Higher Score:**

The reviews concluded that the comparison could be made fairer by reducing experimental setup differences, and reanalyzing the results on the subpopulation shift. Since there are no algorithmic contributions, the rigor around this comparison is critical for acceptance of this work.

**Justification For Why Not Lower Score:**

N/A

---

### Decision · Program_Chairs · 2024-01-16

Reject